# Quantification of *Myxococcus xanthus* Aggregation and Rippling Behaviors: Deep-Learning Transformation of Phase-Contrast into Fluorescence Microscopy Images

**DOI:** 10.3390/microorganisms9091954

**Published:** 2021-09-14

**Authors:** Jiangguo Zhang, Jessica A. Comstock, Christopher R. Cotter, Patrick A. Murphy, Weili Nie, Roy D. Welch, Ankit B. Patel, Oleg A. Igoshin

**Affiliations:** 1Department of Bioengineering, Rice University, Houston, TX 77005, USA; jiangguo.zhang@rice.edu (J.Z.); cotter@sciencesundries.com (C.R.C.); pam12@rice.edu (P.A.M.); 2Department of Biology, Syracuse University, Syracuse, NY 13244, USA; jacomsto@syr.edu (J.A.C.); rowelch@syr.edu (R.D.W.); 3Department of Electrical and Computer Engineering, Rice University, Houston, TX 77005, USA; monde.nie@gmail.com (W.N.); abp4@rice.edu (A.B.P.); 4Department of Neuroscience, Baylor College of Medicine, Houston, TX 77005, USA; 5Departments of Biosciences and of Chemistry, Rice University, Houston, TX 77005, USA

**Keywords:** *Myxococcus xanthus*, phase contrast microscopy, fluorescence microscopy, aggregation, rippling, deep learning, generative adversarial network

## Abstract

*Myxococcus xanthus* bacteria are a model system for understanding pattern formation and collective cell behaviors. When starving, cells aggregate into fruiting bodies to form metabolically inert spores. During predation, cells self-organize into traveling cell-density waves termed ripples. Both phase-contrast and fluorescence microscopy are used to observe these patterns but each has its limitations. Phase-contrast images have higher contrast, but the resulting image intensities lose their correlation with cell density. The intensities of fluorescence microscopy images, on the other hand, are well-correlated with cell density, enabling better segmentation of aggregates and better visualization of streaming patterns in between aggregates; however, fluorescence microscopy requires the engineering of cells to express fluorescent proteins and can be phototoxic to cells. To combine the advantages of both imaging methodologies, we develop a generative adversarial network that converts phase-contrast into synthesized fluorescent images. By including an additional histogram-equalized output to the state-of-the-art pix2pixHD algorithm, our model generates accurate images of aggregates and streams, enabling the estimation of aggregate positions and sizes, but with small shifts of their boundaries. Further training on ripple patterns enables accurate estimation of the rippling wavelength. Our methods are thus applicable for many other phenotypic behaviors and pattern formation studies.

## 1. Introduction

Multicellular self-organization is important for biological processes across all kingdoms of life [1,2,3,4,5]. The development of a complicated living system requires many iterative self-organizing steps, from the creation of tissues and organs, to organisms and interacting communities. In bacteria, self-organization into biofilms can contribute to virulence in addition to providing protection from environmental stressors such as desiccation or antimicrobial agents [6]. For example, developmental self-organization is a crucial part of the myxobacterial stress response, as observed in the formation of starvation-induced fruiting bodies in a soil-dwelling bacterium *Myxococcus xanthus* [7].

*M. xanthus* has long been a model organism for studying collective behaviors [8]. During vegetative growth on a surface, *M. xanthus* exhibits swarming, a multicellular behavior defined by cells migrating together to efficiently colonize a substrate [9]. When a swarm encounters prey bacteria, cells organize into traveling waves of high-cell-density crests separated by lower-density troughs called ripples [10,11,12,13,14]. These waves may allow *M. xanthus* cells to quickly cover their prey and remain in place for longer while lysing prey cells and scavenging the resulting nutrients [14]. Nutrient limitation initiates a third self-organizing behavior, triggering the population of starving cells to aggregate into fruiting bodies where some will differentiate into environmentally resistant spores [7].

Microcinematography, i.e., time-lapse microscopy, is a useful tool for observing these behaviors during the *M. xanthus* life cycle, as time-lapse microscopic imaging is capable of capturing these dynamic biological patterns [11,13,15,16]. Phase-contrast and fluorescence microscopy are commonly used for visualizing live bacterial cells, but each comes with its set of advantages and drawbacks. While phase-contrast microscopy provides enhanced contrast for observing transparent cells on an agar substrate, it can introduce artifacts. Two common artifacts are halos at the boundary of objects [17], and shade-off [18]—gradients in intensity in large features of a specimen—which can potentially obscure aspects of the behaviors of interest. Fluorescent microscopy is useful for observing cells without the background artifacts of conventional bright-field or phase-contrast microscopy, and it provides a way to estimate the local density of bacterial populations [15]. However, the time demands of introducing genes that encode fluorescent proteins to existing mutant libraries and strain collections are often prohibitive to conducting large-scale analyses. Moreover, the additional light exposure required for fluorescent imaging can be phototoxic and, therefore, can lead to cell behavior changes and/or limit the observation duration [19].

One way to avoid the disadvantages of these imaging techniques is to develop an image-processing algorithm that transforms phase-contrast microscopy images into synthesized fluorescent images. If successful, such an algorithm could allow researchers to take advantage of the wealth of information contained in fluorescent images without the associated time constraints and complications of fluorescence microscopy. Here, we hypothesized that recent exciting advances in the field of deep learning for image processing [20,21] can be leveraged to develop and train such an algorithm. In particular, image-to-image translation has been used for tasks such as image colorization [22], image denoising [23], semantic synthesis [24], style transfer [25], etc. These applications have mostly employed Generative Adversarial Networks (GANs) [26]. The GAN model consists of two interacting neural networks, a generator and a discriminator, which are trained jointly by playing a zero-sum game. The generator tries to synthesize images to fool the discriminator, while the discriminator tries to distinguish real samples from synthesized ones. Although early GAN models suffered from low resolution [24] and a lack of detail and realistic textures [27], recently, the quality of image-to-image translation has significantly improved due to the use of more advanced conditional GAN models, such as pix2pixHD [28], which can generate high-resolution images, and SPADE [29], which can control the desired look of the output with a style image input. However, these models have never been applied to synthesize high-resolution microscopic images.

In this paper, we developed such an image-to-image transformation algorithm and assessed its performance in terms of quantification of aggregation and rippling behaviors. In particular, we prepared phase-contrast and fluorescent time-lapse movies of the aggregation and rippling phenotypes of *M. xanthus* cells labeled with tdTomato fluorescent protein, and compared the detection and segmentation of biological features in both sets of images. Based on the state-of-the-art pix2pixHD model, we then developed a new conditional GAN network architecture called pix2pixHD-HE that can be trained on phase-contrast and fluorescent images taken of the same field of view at the same time. We applied our model to transform phase-contrast images into synthesized fluorescent images. To assess the model’s performance, we compared the image quality with the pix2pixHD baseline. We also compared the position and size of aggregates segmented from our synthesized images to those segmented from real fluorescent images. To determine whether the model is applicable to other self-organization patterns, we applied the model to time-lapse movies of rippling and used it to quantify the ripple wavelength. The results demonstrate the generalizability of our approach for a wide range of cell patterns, but some training could be required for new phenomena.

## 2. Materials and Methods

### 2.1. Strains and Culture Conditions

*M. xanthus* tdTomato-expressing strain LS3908 [15] and GFP-expressing strain DK10547 [11], and *E. coli* K12 were used in this study. *M. xanthus* strains were grown overnight at 32 ${}^{\circ }$C with vigorous shaking in CTTYE broth (1% Casein Peptone (Remel, San Diego, CA, USA), 0.5% Bacto Yeast Extract (BD Biosciences, Franklin Lakes, NJ, USA), 10 mM Tris (pH 8.0), 1 mM KH(H${}_{2}$)PO${}_{4}$, 8 mM MgSO${}_{4}$) supplemented with 10 μg/mL oxytetracycline and 1 mM isopropyl β-D-1-thiogalactopyrano-side (IPTG) for LS3908 or with 40 μg/mL kanamycin for DK10547. For development assays, mid-log phase cells were harvested, resuspended in TPM starvation buffer (10 mM Tris (pH 7.6), 1 mM KH(H${}_{2}$)PO${}_{4}$, 8 mM MgSO${}_{4}$) to a concentration of $5\times 10^{9}$ cells/mL (or $1\times 10^{10}$ cells/mL for high-density and $2.5\times 10^{9}$ cells/mL for low-density samples) and plated on a microscope slide chamber prepared, as previously described [30], containing 1% agarose TPM media with 1 mM IPTG added. To track cells in streams during development, LS3908 cells were diluted 1:800 into DK10547 and plated on a microscope slide chamber as above.

To induce rippling, *E. coli* K12 cells were grown overnight in LB broth (Sigma, St. Louis, MO, USA) in a 37 ${}^{\circ }$C incubator with vigorous shaking, harvested and washed in TPM buffer, and plated on 1% or 0.6% agarose microscope slide chambers containing TPM supplemented with 1 mM IPTG. Once cell spots of *E. coli* were dry, LS3908 cells from an overnight culture were prepared in TPM as above, 3 μL were plated in the center of the *E. coli* spot, and the slide was incubated in the dark at 32 ${}^{\circ }$C for 8–10 h before imaging to provide time for rippling to initiate.

### 2.2. Time-Lapse Imaging

Microscope slide chambers were placed on a stage warmer (20/20 Technologies, Wilmington, NC, USA) set to 32 ${}^{\circ }$C on a Nikon Eclipse E-400 microscope (Nikon Instruments, Melville, NY, USA). A pco.panda 4.2 sCMOS camera and NIS-Elements software were used for automated time-lapse imaging, capturing a phase contrast and fluorescent image every 60 s for a total of 24 h for development movies and 8 h for rippling movies. Phase-contrast images were taken with 70 ms exposure, and transmitted light was manually shuttered with a Uniblitz VMM-D1 shutter (Uniblitz Electronics, Rochester, NY, USA) when not actively imaging. Fluorescent tdTomato-expressing samples were imaged with 400 ms exposure with a Sola LED light source (Lumencore) at 75% intensity, and GFP-expressing samples were imaged with 200 ms exposure at 35% intensity. A MAC6000 system filter wheel controller and focus control module (Ludl Electronic Products, Ltd., Hawthorne, NY, USA) were used for control of the fluorescent filter wheel and the autofocus feature.

### 2.3. Image Processing

The raw images captured by phase-contrast and fluorescence microscopes have inconsistent contrast and we do not need such high-resolution images to resolve aggregates and streams. Furthermore, using original resolution for training may be too slow and require too much GPU RAM memory. Therefore, we scaled down and cropped the images to train an efficient model. The processing pipeline is shown in Figure 1. For detailed methods, see Appendix A.

### 2.4. Cell Tracking

Cell tracking was performed as described by Cotter et al. [15] for 15 min. The trajectories of tracked cells are superimposed on phase-contrast, fluorescence microscopy, and histogram-equalized fluorescence microscopy images at the initial frame of tracking.

### 2.5. Image Segmentation

The image segmentation algorithm converts grayscale images to binary images, where the foreground represents aggregates and the background represents the interaggregate space (Results Section 3.1). Segmentation is based on the observation that aggregates are brighter in intensity than the surrounding field and consists of two steps. First, we applied a bandpass filter as described in [15] to the input images to filter out low-frequency background and high-frequency noise. Second, we used Otsu’s method [31] to perform automatic image thresholding. The algorithm returned a threshold separating the pixels into foreground and background, where the interclass difference is maximized. Finally, we applied erosion and dilation to remove small foreground spots, fill the holes in the foreground, and smooth the foreground boundaries. The detailed method is described in Appendix A.

### 2.6. Network Architectures and Learning Algorithm

We adapted our generator and discriminator architectures from those in [28]. We modified the pix2pixHD framework by adding a histogram-equalized output channel in the generator (Figure 2). The pix2pixHD-HE generator network architecture is illustrated in Figure 2. We searched through different branch points in the generator architecture (Appendix A) for the best performing network architecture as measured by MSE and SSIM, while keeping other hyperparameters the same with pix2pixHD. All the networks were trained from scratch with Adam optimizer and a learning rate of $5\times 10^{-5}$ for 2000 epochs. We kept the loss weight of G2 the same and linearly decayed the loss weight of G1 for the first 500 epochs. Details of the architectural parameters and hyperparameters are provided in Appendix A.

### 2.7. Evaluation Metrics

#### 2.7.1. Mean Square Error (MSE) and Structural Similarity Index Measure (SSIM)

To evaluate the similarity between synthesized images and real fluorescent images, we computed and compared two metrics—(i) the Mean Squared Error (MSE) and (ii) the Structural Similarity Index Measure (SSIM)—between the predicted and target images. We did not use another commonly used measure, peak signal-to-noise ratio, as it is mathematically equivalent to MSE when using normalized images. Note that unlike many image reconstruction tasks in which only high-level features need to be accurately recovered, our case requires capturing more low-level features and ideally needs to recover how cell density changes in space and time. Given that cell density is well-correlated to fluorescence microscopy image intensity [15], we need the exact match of the intensity distribution of the fluorescence image to recover biologically relevant features such as position and relative density of the aggregates and streams. Therefore, we used both MSE and SSIM as our metrics on both normalized images and histogram-equalized images as they capture pixel-to-pixel variation on different levels.

For two images represented by intensity value matrices *r* (real fluorescent image) and *s* (synthesized fluorescent image), the MSE is defined as
(1)$$MSE\left(r,s\right)=\frac{\sum_{w}^{N_{w}}\sum_{l}^{N_{l}}{\left(r_{l,w}-s_{l,w}\right)}^{2}}{N_{l}N_{w}},$$
where $N_{l}$ is the image length and $N_{w}$ is the image width. For images with intensity scaled to the range [−1,1], the MSE is in the range [0,4]. The minimal value of 0 is achieved when two images are identical for all pixels. To give the MSE a more intuitive interpretation, we applied spatial perturbations and Gaussian noise on real fluorescent images and calculated when these perturbations achieved the same MSE as between real and generated images. These comparisons are described in detail in Appendix A.

The SSIM is a common alternative metric to the MSE, and captures information about differences between the luminance, contrast, and structure of two images. We did not use it to train the network, but used it solely as a comparison metric for real and synthesized images. It is defined as
(2)$$SSIM\left(r,s\right)=\frac{2\mu_{r}\mu_{s}+c_{1}}{\mu_{r}^{2}+\mu_{s}^{2}+c_{1}}\frac{2\sigma_{rs}+c_{2}}{\sigma_{r}^{2}+\sigma_{s}^{2}+c_{2}},$$
where $\mu_{r}$, $\mu_{s}$ are the mean pixel intensities of two images, $\sigma_{r}^{2}$ and $\sigma_{s}^{2}$ are the corresponding variances, and $\sigma_{rs}$ is the covariance between the two images. $c_{1}$ and $c_{2}$ are constants to avoid zero in the denominators, and the values are related to the data range. Here, $c_{1}=0.004$ and $c_{2}=0.0036$ by default for images with intensity range [−1,1]. The maximum value of SSIM is 1 when two images are identical for all pixels, and the minimal value is 0.

#### 2.7.2. Aggregate Comparison

To compare the segmented aggregates, we first matched the aggregates segmented from different images. The positions of the aggregate centroids were compared; then, the aggregates were matched based on their overlap and distance. When comparing the two images, pairs of aggregates—one from each image with the shortest distance between their centroids—are considered the same aggregate unless they have no overlapping pixels. If there is no overlapping region, or if the aggregate segmented from one image is twice as large as the other one, the aggregate pair is considered a mismatch. The distribution of distances between centroid positions and the area ratio of the matched aggregate pairs are plotted in histograms (Results Section 3.5).

Then, we compared the aggregate segmentation on a pixel level. We used precision and recall to show the segmentation accuracy, assuming segmentation of real fluorescent images to be ground truth. In that case, we labeled the segmented image with four labels (Results Section 3.5)—True positives (TP, white) are the pixels correctly identified as aggregate pixels in the generated images. False positives (FP, orange) are the pixels incorrectly identified as aggregate pixels. True negatives (TN, black) are the pixels correctly identified as interaggregate pixels. False negatives (FN, red) are the pixels incorrectly identified as interaggregate pixels. The precision is defined as the fraction of pixels in the aggregates extracted from the generated images that were correct, i.e.,
(3)$$\mathrm{precision}=\frac{\mathrm{TP}}{\mathrm{TP}+\mathrm{FP}}$$

On the other hand, recall is a fraction of aggregate pixels that are correctly extracted from generated image, i.e.,
(4)$$\mathrm{recall}=\frac{\mathrm{TP}}{\mathrm{TP}+\mathrm{FN}}$$

#### 2.7.3. Rippling Wavelength Detection

To determine the ripple wavelength in an L×L rippling image, we performed a two-dimensional Discrete Fourier Transformation (DFT) in polar coordinates. To simplify the problem, we first performed a 2-D DFT in Cartesian coordinates $\left(x,y\right)\to \left(k_{x},k_{y}\right)$ with the *scipy* package [32]; then, we transformed the absolute values to polar coordinates $\left(k_{x},k_{y}\right)\to \left(k,\theta \right)$.

We then calculated the amplitudes for different integer-valued radii *k* by averaging the amplitudes across all angles θ in the radius interval [k,k+1). We smoothed the amplitudes by calculating the moving average with window size 3 to remove the small fluctuations. The wave-number $k_{1}$ is determined as the peak value above a certain frequency, which was set to 6 in our 504×504 image. A lower-frequency corresponds to gradual changes in the image intensity; the first peak value $k_{0}$ would result in a wavelength above 84 μm, corresponding not to rippling but to large-scale features. For the majority of the images, the peak corresponds to the values of $k_{1}$. Finally, the wavelength $\lambda_{k}$ was calculated with
(5)$$\lambda_{k}=\frac{L}{k_{1}}$$

The sample rippling wavelength calculation is shown in Appendix A.

## 3. Results

### 3.1. Fluorescent Images Outperform Phase-Contrast Images in Aggregate Segmentation

We collected a total of 14 time-lapse movies taken from the onset of starvation through the first 24 h of *M. xanthus* development at varying densities (see Appendix A for more details). Across all replicates, aggregates formed at an average of 7.7 ± 1.2 h, meaning that the data includes preaggregation cell patterns; patterns that occur during aggregate formation; and finally, patterns seen after aggregates are stable. To compare phase-contrast and fluorescence microscopy images of aggregation, we recorded the same field of view with both methods taking simultaneous snapshots every minute over 24 h of development. In order to correct for the possibility of unequal contrast and illumination in the course of the observation, we normalized the image intensity on both channels for each frame with Contrast Limited Adaptive Histogram Equalization (CLAHE). An example image of phase contrast microscopy is shown in Figure 3A and the corresponding fluorescence microscopy image in Figure 3B. Appendix A shows both channels side-by-side over the course of the aggregation. Clearly, the grayscale intensity of the phase-contrast images does not directly correlate with underlying cell density. The aggregates are on average much brighter than the background but all larger aggregates contain dark bands or spots. In contrast, fluorescent images show aggregates with a more uniform intensity that are clearly brighter than the background such as in Figure 3B; moreover, intermediate intensity patterns connecting the aggregates can be seen, suggesting that these are the streams along which cells move to the aggregates or in-between the aggregates.

To further qualitatively compare two imaging modalities in their ability to extract aggregates, we adapted and fine-tuned image segmentation algorithms previously described by Cotter et al. [15] (see Methods Section 2.5 for more details). We first performed aggregate segmentation on phase-contrast images. The stable aggregate boundaries are well-defined by high-intensity pixels, and the holes in the middle are filled with our algorithm. However, for some immature (i.e., unstable) aggregates, the aggregate boundaries are not well-defined, and dark bands or spots in the middle make them difficult to distinguish from interaggregate spaces (red rectangles in Figure 3C). These aggregates are often distorted or missing when segmented from phase-contrast images. Applying the same image segmentation methodology to fluorescent images, we obtained the mature aggregates without holes. The immature aggregates are also clearly segmented (Figure 3D). Therefore, fluorescent images outperform phase-contrast images in aggregate segmentation.

### 3.2. Histogram Equalization (HE) Helps Visualize Streams on Fluorescent Images

To further characterize both image modalities on their ability to depict the patterns of cell motion in the interaggregate spaces, we decided to compare the patterns in image intensity with the cell trajectories. To this end, we designed an experiment to track how cells move in the stream regions by labeling a small (1:800) fraction of cells with tdTomato and the rest of the cells with GFP. We then tracked cells for 15 min (see Methods Section 2.4 for details) and superimposed these tracks onto the initial frame in each imaging modality to see if the interaggregate patterns indeed correspond to streams. As shown in Figure 4A, in the CLAHE normalized fluorescent images, cell trajectories (red) indeed somewhat align with the regions of intermediate intensity between aggregates. However, the interaggregate spacing has lower contrast than the aggregates, so the streams are not clear. To enhance the pixel contrast outside aggregates, we chose a different normalization, the so-called Histogram Equalization (HE) (Figure 4B). This normalization allows for areas of lower local contrast to gain a higher contrast. The streams on HE images are greatly enhanced and have higher intensity values than the surrounding background. As expected, most cell trajectories overlap with the high-intensity streams and the aggregates (Figure 4B).

Phase-contrast images, with (Figure 4C) or without (Figure 4D) histogram equalization, have pixel intensity values under the cell trajectories that are neither higher nor lower than the surrounding regions. This means that the cell streams are undetectable in phase-contrast images. Thus, the patterns of cell motility in the interaggregate space are also better represented in fluorescent images than in phase-contrast images.

### 3.3. Deep Neural Network Synthesizes Fluorescent Images with Aggregates and Streams

Although fluorescence microscopy is better for quantifying both the aggregates and streams in the interaggregate space, it requires engineering cells to express fluorescent proteins and may lead to cell damage due to phototoxicity [19]. To combine the advantages of both imaging methodologies, we aimed to design and train an image style transfer model to convert phase-contrast to fluorescence microscopy images. A recently developed pix2pixHD model [28] (Appendix A) has been shown to be trainable for high-resolution image transformation of semantic synthesis and sketch-to-image synthesis. We therefore first attempted to use pairs of corresponding phase-contrast and tdTomato fluorescent images as respective inputs and outputs for model training. We trained the model on 1020 images from 7 movies of $5\times 10^{9}$ cells/mL and 434 images from 3 movies of $2.5\times 10^{9}$ cells/mL for 2000 epochs, because a larger training set can improve the model accuracy [33], and all images in our training set have similar sizes of aggregates (Appendix A). For validation and testing, we applied the model to images of experimental replicates with 3 different cell densities ($2.5\times 10^{9},5\times 10^{9},1\times 10^{10}$ cells/mL. The results are illustrated in Figure 5 (Appendix A) with synthesized images (Figure 5C and Appendix A) looking somewhat similar to the original fluorescent images (Figure 5B and Appendix A). Visually, it appears that the pix2pixHD model successfully recovers the majority of aggregates and some patterns in the interaggregate regions.

We further checked the model performance in the interaggregate space by applying histogram equalization to the synthesized image (Figure 5F and Appendix A) and comparing it with the histogram-equalized real fluorescent image (Figure 5E and Appendix A). Despite similarities in some regions, the synthesized image is blurry in areas (e.g., see the area indicated by the red box in Figure 5E) where no streaming patters are visible, or the image has uneven contrast (e.g., see the bottom region in Appendix A). The result shows that the standard pix2pixHD model is not able to learn the low-intensity interaggregate streaming patterns.

To help the model learn streaming patterns, we changed the model architecture to include histogram-equalized fluorescent images as an extra output channel. It is necessary because histogram equalization is not an invertible process. The original image cannot be derived from the histogram-equalized image. Therefore, the resulting architecture, which we call the *pix2pixHD-HE* model (Figure 2), learns aggregate patterns from the normalized fluorescent images and interaggregate patterns from the histogram-equalized fluorescent images. We trained the model with the same training set used for the *pix2pixHD* model for 2000 epochs, and tested it on the same experimental replicate test set. The results shown in Figure 5D,G; Appendix A demonstrate that the new model recovers both aggregate and streaming features better than pix2pixHD.

### 3.4. The Synthesized Images Show Good Global Agreement with the Real Fluorescent Images

When evaluating the synthesized image quality, we seek to get a good agreement between the image intensity and the corresponding cell density throughout the image. Therefore, we employed MSE and SSIM, image similarity evaluation measures that compare images on a pixel-by-pixel basis. The details are presented in Table 1 and Table 2. We first applied our model on images of experimental replicates with cell densities of $2.5\times 10^{9}$ and $5\times 10^{9}$ cells/mL. The pix2pixHD-HE performs equally to pix2pixHD on both training sets and test sets, with small differences.

To check the image synthesize performance in detail, we applied the two metrics on 1440 images taken every minute in a 24-h training set, and compared the metrics for both pix2pixHD-HE and pix2pixHD across the whole movie (Appendix A). From 500 to 1200 min, our pix2pixHD-HE model performed significantly better than pix2pixHD. However, outside of this time interval, the performance of pix2pixHD-HE has no significant difference from pix2pixHD. To understand why, we examined the images taken at the 500 min, 800 min, and 1200 min time points. The 500 min image corresponds to the time when the aggregates start to form—the interaggregate space is changing rapidly, and the streams are changing their location (Appendix A). By the 800 min mark, the aggregates are mostly formed and starting to mature, and the streams connecting them are forming stable spatial patterns (Appendix A). At 1200 min, most of the aggregates are mature and few cells remain in the streams. Thus, the streams connecting different aggregates are dimmer in their fluorescence intensity (Appendix A). The full dynamics of the aggregation of the real and synthesized images over 24 h is shown in Appendix A. We conclude that pix2pixHD-HE outperforms pix2pixHD with both MSE and SSIM metrics only in the time period between 500 to 1200 min, when both aggregates and streams are present. This result suggests that the branched histogram-equalized output may assists the pix2pixHD-HE to learn the stream transformation. Notably, this time period slightly varies for different cell densities and biological replicates; nevertheless, each movie has a similar time period in which pix2pixHD-HE can better transform images than pix2pixHD.

To better quantify the difference between real and synthesized images, we calculated both a spatial displacement shift in pixels and a level of added Gaussian noise in the pixel intensity that generated equivalent differences in the values of the MSE and SSIM. For a given real image, we calculate the smallest spatial shift such that the MSE between the real and shifted image exceeds the MSE between the real and generated images. The results in pixels can be computed for every frame of the time-lapse movie; they are averaged and then converted to a single value in μm (Table 1 and Table 2). A similar approach holds for the SSIM. In the same fashion, we can define a minimum standard deviation for added Gaussian noise—given as a percentage of the original intensity scale—that when added to each pixel will also give the same MSE. Again, averaging these percentages over the duration of the movie allows us to get a better interpretation of the pixel-by-pixel conversion accuracy.

We evaluated our model’s performance on the images of the experimental replicates. Without histogram equalization (Table 1) on the output image, the comparable spatial shifts for the MSE are around 7.0 microns, while the relative Gaussian noise σ is around 7.8%. With SSIM, the comparable spatial differences are around 2.1 microns. After histogram equalization (Table 2), the comparable spatial differences for MSE are around 3.6 microns, while the relative Gaussian noise σ is around 19.7%. With SSIM, the comparable spatial differences are around 2.3 microns. These errors are roughly equivalent to the average cell length during development (about 7 microns [34]).

We further tested the generalizability of our model by applying it to images taken under different experimental conditions. Specifically, we tested whether our model could, without additional training, generate accurate synthesized fluorescent images of *M. xanthus* aggregation at a 2-fold-higher cell density, where the observable patterns are slightly different. Doubling the cells in the field of view leads to a proportional increase in dead cells and other debris that can impact the quality of ground truth fluorescent images (Appendix A). However, these dark regions of dead cells and debris on the fluorescent images disappear as cells move over the agar to form initial aggregates, so we consider only images captured after 8 h for this analysis. The size of aggregates is generally larger in these images (Appendix A), which leads to larger MSE and smaller SSIM. The comparable spatial shifts are 22.16 and 6.171 microns. This result shows that we are able to generate images for higher cell densities even with training only on images of lower cell densities.

### 3.5. The Aggregates Segmented from the Synthesized Images Show Good Agreement in Their Positions and Sizes but Some Variability in Aggregate Boundaries

To evaluate the aggregate segmentation in the synthesized images, we applied the same image segmentation method on both the synthesized and the original phase-contrast images from 600 min to 1440 min, when the aggregates in three test sets were formed. We then compared both the centroid positions and sizes of corresponding aggregates between the real fluorescent images and both the synthesized and phase-contrast images. Figure 6A shows the centroid displacement from the real images for both synthesized images (blue) and phase-contrast images (orange) at a cell density of $1\times 10^{10}$ cells/mL. It is clear that the displacement of aggregates on synthesized images are smaller than that on phase-contrast images, indicating that the synthesized images more faithfully reproduce aggregate position. For images taken at a cell density of $5\times 10^{9}$ cells/mL, the results are listed in Table 3. The displacement of aggregates segmented from synthesized images is greater than at high density, but it is still lower than the displacement for the lower-density phase-contrast images.

We next performed an analysis on the relative area of segmented aggregates compared with the aggregates in real fluorescent images. Figure 6B shows the relative area of aggregates in synthesized images (blue) and phase-contrast images (orange) for cell density $1\times 10^{10}$ cells/mL compared with real fluorescent images. The area of aggregates segmented from synthesized images is close to that segmented from real fluorescent images (Table 3). However, the area of aggregates segmented from phase-contrast images is smaller than those segmented from real fluorescent images, indicating that the segmentation applied to phase contrast loses parts of the aggregates. For images at a cell density of $5\times 10^{9}$ cells/mL, the area of aggregates segmented from synthesized images is also close to that segmented from real fluorescent images. Again, however, the area of aggregates segmented from phase-contrast images is smaller than that segmented from real fluorescent images, albeit to a lesser extent.

The global accuracy of aggregates segmented from synthesized images was evaluated on a pixel-by-pixel basis by measuring precision (Equation (Equation 3)) and recall (Equation (Equation 4)). Precision is the fraction of correctly segmented aggregate pixels among the aggregate pixels segmented in synthesized images, whereas recall is the fraction of aggregate pixels segmented in real fluorescent images that were retrieved in synthesized images. Under both $5\times 10^{9}$ and $1\times 10^{10}$ cells/mL conditions, the aggregates in synthesized images have higher precision and recall than those in phase-contrast images (Figure 6B and Table 3).

On comparing the results of synthesized images with different cell densities (Table 3), we found that our model performs better on high-cell-density images. When visualizing the aggregates extracted from high-density images (Figure 6), we found that the aggregates are larger in high-density images. As most of the distortion during synthesized image generation happens on the aggregate boundaries, larger aggregates will have a smaller displacement and relative area compared with the original. Appendix A show the aggregate segmentation results for different cell densities and indicate that our model has higher precision and recall on large aggregates.

### 3.6. With Further Training, the Model Can Be Used to Convert Phase-Contrast Images of Rippling and Estimate Its Wavelength

Finally, we investigated if our model can accurately generate other self-organization patterns of *M. xanthus*. To this end, we prepared *M. xanthus* cells together with *E. coli* prey cells to induce rippling. We then recorded the same field of view with the phase-contrast and fluorescence microscopy, taking a snapshot every 60 s over 4 to 8 h while rippling occurred. When comparing the phase-contrast (Figure 7A) and fluorescence microscopy (Figure 7B) images of rippling, one can readily see that the wave-crests are much more apparent in the fluorescent images. For example, in the upper-left region of the phase-contrast image (red box in Figure 7A), the waves cannot be visibly detected, despite the clear presence of ripples in the corresponding fluorescent image. Therefore, we hypothesized that ripple wavelength would not be as accurately quantified from the phase-contrast images. To verify this, we computed the Fourier transform spectrum for each image and used the peak value above a certain threshold in the spectra to compute rippling wavelength (see Methods Section 2.7.3 for the details). The wavelength obtained from a strong peak in the spectrum for the fluorescence images ($\lambda_{k}∼46$
μm) matches the distance between neighboring crests. For phase-contrast images, the wavelength estimated from the whole image is the same ($\lambda_{k}∼46$
μm). However, in the upper-left region, the peak in the spectrum is not very strong and the wavelength corresponding to it ($\lambda_{k}∼21$
μm) is too small to match the observed ripple patterns, demonstrating that the wavelength estimation is much more accurate when quantified from fluorescent images. Thus, we test whether the rippling patterns and wavelength estimates we observe using our image transformation from phase-contrast to effective fluorescence images match those observed in the ground truth fluorescence microscopy images.

First, we apply our model trained on the aggregation patterns to rippling images without extra training. As one can see in Figure 7C, the rippling pattern is not visible in some regions of the synthesized image and the patterns are hardly more apparent in comparison to the input phase-contrast image. Not surprisingly, the wavelength estimate for that image was also low ($\lambda_{k}∼21$
μm). Next, we wanted to test if we can train the pix2pixHD-HE to learn the rippling patterns. However, it takes a significant amount of time to train a new model from scratch, and we did not have as many rippling movies as aggregate movies. We decided to use a training technique called transfer learning that can use a smaller training set and save training time.

Since we are still transforming *M. xanthus* images, just for a different experimental condition, the knowledge gained by our model to transform phase-contrast images to synthesized fluorescent images can be reused. We performed this knowledge transfer by using mapping-based deep transfer learning [35]. The previous model parameters are taken as the start point for training with the updated data set. We incorporated 731 rippling images from 2 movies in the training set and repeated training for 500 epochs. Compared with training from scratch again with 2000 epochs, this is a great increase in efficiency. This indicates that the model learned the ripple patterns based on its previous knowledge of aggregate patterns, resulting in an overall decrease in training time. The resulting model applied to Figure 7A (from a separate biological replicate, not present in the training set) shows a rippling pattern with intensity and contrast resembling that of the real fluorescent image (Figure 7B,D) over the entire field of view. The wavelength detected in the top-left region is $\lambda_{k}∼46$
μm, which exactly matches the real fluorescent image. When we calculate the wavelengths from different images across a 4-hour movie (Appendix A), the wavelengths calculated from images synthesized by the trained model are close to those calculated from real fluorescent images. On the contrary, the wavelengths in phase-contrast images are lower, and the wavelengths for the untrained model fluctuate across different frames. The result shows that our model has learned the hidden rippling features after training.

## 4. Discussion

When choosing imaging modalities for observing live bacterial cells, it is important to consider that each one may be best at capturing a particular biological feature, but could distort or obscure other equally important features. For example, phase-contrast microscopy accentuates the contrast of nearly transparent *M. xanthus* cells to improve visualization of swarm-level patterns during aggregate development, but here, we show that the halo and shade-off effects slightly distort the image and misrepresent information about aggregate size and position (Figure 6B, Table 3) as well as the exact position of cells with respect to interaggregate streams (Figure 4). Fluorescence microscopy does not distort images in the same way, and is, therefore, better at representing aggregate features such as size and position, but at a cost; it requires a genetic mutation to introduce and express the fluorescent protein in *M. xanthus*, and an excitation light source to detect the protein in the cells. We sought to overcome the limitations of both imaging modalities by manipulating phase-contrast image data so that they could be directly compared to fluorescent image data.

To accomplish this, we built a high-resolution image transformation conditional GAN model, pix2pixHD-HE, to transform phase-contrast images to synthesized fluorescent images of *M. xanthus*. Using traditional evaluation metrics for global agreement between images—Mean Square Error (MSE) and Structural Similarity Index Measure (SSIM)—we determined that our model performs better than the state-of-the-art model pix2pixHD during aggregate formation (Figure 5). Furthermore, aggregates segmented from the converted images show excellent agreement in their position and size when compared with those segmented from real fluorescent images. Notably, with training on images of lower cell densities ($2.5\times 10^{9}$ and $5\times 10^{9}$ cells/mL), the model can synthesize images of aggregation at high cell density ($1\times 10^{10}$ cells/mL) with small error (Table 1 and Table 2). With additional training, our model can also generate other multicellular dynamic patterns associated with *M. xanthus* swarms, as observed in the generation of synthesized fluorescent images of rippling that allow for accurate quantification of ripple wavelength (Figure 7D).

Fluorescence microscopes were invented to overcome the lower resolving power of ordinary optical microscopes. Although fluorescent images can better represent live-cell positions, the associated phototoxicity is an unavoidable problem that can affect cell behavior. The exposure to excitation light may change the morphology of the observed specimen [19] or could influence the behavior of bacterial cells tracked over time. Methods to reduce the impact of phototoxicity include limiting illumination to the focal plane, reducing the peak intensity, pulsing the illumination, and using longer wavelengths [19]. However, many of these methods can require specialized imaging equipment, and none of them eliminate exposure of the sample to excitation illumination altogether. Further, strains must be engineered to express the desired fluorophore, which may alter cell behavior and typically precludes high-throughput analyses involving fluorescence imaging. Our suggested approach to avoid phototoxic effects is to bypass fluorescence microscopy entirely and synthesize the fluorescent image from other nonfluorescence microscopic images, thereby eliminating excitation light exposure while retaining the ability to quantify biological patterns. Our model transforms phase-contrast images into synthesized fluorescent images, providing better resolution for observing cell behaviors while avoiding phototoxic effects.

While our model is not broadly applicable to every situation where fluorescence microscopy is required, it is particularly useful in providing a way to quantify phenotypes and behaviors that can be observed, but not accurately measured, in phase-contrast images. Our experiments with *M. xanthus* demonstrate that our model is useful in instances where an entire population of cells would be labeled with a fluorophore, as we would be able to accurately extract information about aggregate position and morphology. While phase-contrast images can approximate these data, there are times when having more accurate data on aggregate size, shape, and location is crucial, and it might be detrimental to quantify these from phase-contrast images. *M. xanthus* development is dynamic, and the factors that cause fruiting bodies to form, grow, shrink, merge, and disappear are not well understood. With training on additional data, our model could improve these investigations by synthesizing fluorescent images of wild-type and/or genetic mutant strains, without requiring the expression of a fluorophore, and quantifying aggregation patterns to determine how the mutations affect aggregation dynamics.

Additionally, in experiments where live bacterial cells are tracked, it is useful to label the subpopulation of tracked cells with one fluorophore and the rest of the cells with another so that the cell paths can be understood in the context of local cell densities and location within the swarm [15]. As shown in Figure 4C,D, cell trajectories can be superimposed over phase-contrast images, but the positions and orientations of those cells relative to streams are missing. Prior research suggests that streams may be part of fruiting body formation and growth [36]; cell behavior in streams is different than that of cells outside of streams or inside fruiting bodies [37]; and due to slime trail following [38], streams may be the avenue by which most cells enter a fruiting body. As such, our pix2pixHD-HE model could be important for improving stream visualization in cell-tracking experiments. A single fluorophore and the associated excitation illumination would still be required to track individual cells, but our model would eliminate the need for another fluorophore, and imaging of background cells could instead be done in phase-contrast and then transformed to generate a synthesized fluorescent image containing both aggregate and stream information.

In addition to the improved visualization of the features discussed above, there are some aspects of *M. xanthus* self-organizing patterns that we cannot observe with the naked eye in phase-contrast images. However, the information is still contained within these images, as our model can transform them into synthesized fluorescent images that accurately depict these patterns. For example, we found that the phase-contrast images do not show immature aggregates well, but that these aggregates are present in the real fluorescent images. (Figure 6A,C). Similarly, the rippling wavelength is sometimes undetectable with phase-contrast (Figure 7A), and when checking spatial frequencies in Fourier space, the rippling frequency is masked by the frequency of the inherent noise (Appendix A). When we transform these phase-contrast images, the position and size of immature aggregates and the wavelength of ripples match the data extracted from the real fluorescent images (reference figures/tables). In this way, we can visualize and quantify features that were undetectable in the phase-contrast images.

The fact that the model was trained on lower cell densities and successfully recovered aggregation patterns at higher cell densities without additional training on the high-density datasets indicates that our model is likely capable of capturing the variation we see in different biological replicates under the same experimental conditions. Aggregate size and count, for example, vary from replicate to replicate, just as they vary from high to low cell densities, and we were capable of recovering them despite this variation. With additional training on rippling datasets, we were also able to use our model to recover synthesized fluorescent images of rippling from phase-contrast images without losing the ability to recover aggregates, highlighting the potential for our model to recognize and quantify multiple different biological patterns. An important caveat is that training should occur over a range of phenotypes for a specific biological pattern. Thus, if we want to use the model to transform images containing new cell patterns, it is necessary to train the model on images of the same phenotype. The specificity of the model could also be useful; however, if a synthesized image appears unrealistic, it is possible that the original image contains a pattern not present in the training set that could be biologically important but is difficult to detect just by looking at the phase-contrast images.

In this paper, we have shown an example of training a rippling image transformation model based on the aggregate image transformation model. If we want to perform a similar image transformation task for a new experimental condition, where we have fewer experimental samples, the pretrained model can be a good starting point to reuse the knowledge learned from previous training. A new model can be trained from the old using transfer learning, as we did with regards to rippling images. In other cases, we may want to extract features other than aggregates and streams. In our case, we had two outputs from pix2pixHD-HE: the normalized fluorescent images and the histogram-equalized fluorescent images. These outputs can be modified as necessary to fit new cases. For example, we could instead synthesize fluorescent images labeled with corresponding fluorophores, each used to capture different features of the bacterial dynamics. Transfer learning would allow us to use the old model wholly or in part to train a new neural network with these two new outputs. These applications of transfer learning may not only generalize our model to different *M. xanthus* phenotypes, but also to different applications within biology and medicine. The simplicity of the model training procedure indicates that it can be easily transferred to other cell systems, such as *Bacillus subtilis*, and *Escherichia coli*, bacterial model systems where self-organizing biological patterns are also important. Furthermore, medical images, such as ultrasound imaging, magnetic resonance imaging (MRI), radiographic imaging, and Cryogenic electron microscopy (Cryo-EM) imaging may also be transferred from one to another using the method detailed in this paper.

## Figures and Tables

**Figure 1 microorganisms-09-01954-f001:**
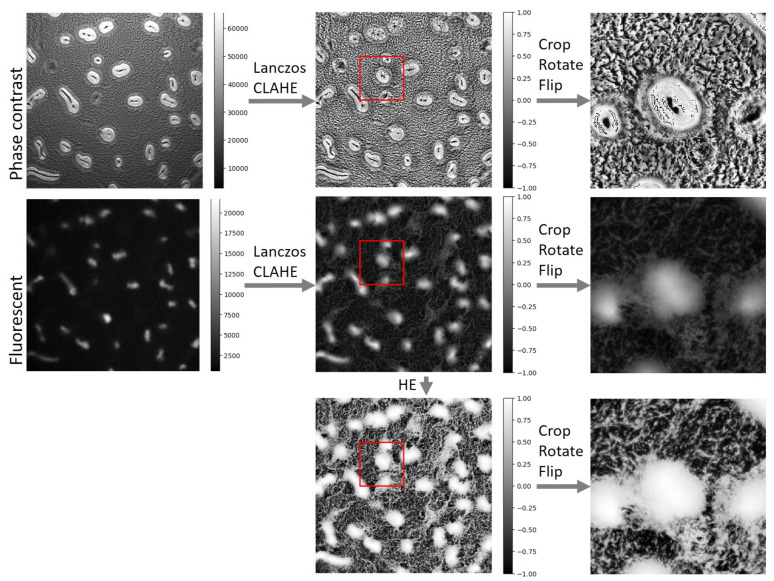
Image-processing pipeline. The intensity ranges are shown in the color bars, and the cropped regions are shown in red boxes. Phase contrast and fluorescent images are from the same experimental replicate and time point. LANCZOS—The Lanczos algorithm; CLAHE—Contrast Limited Adaptive Histogram Equalization.

**Figure 2 microorganisms-09-01954-f002:**
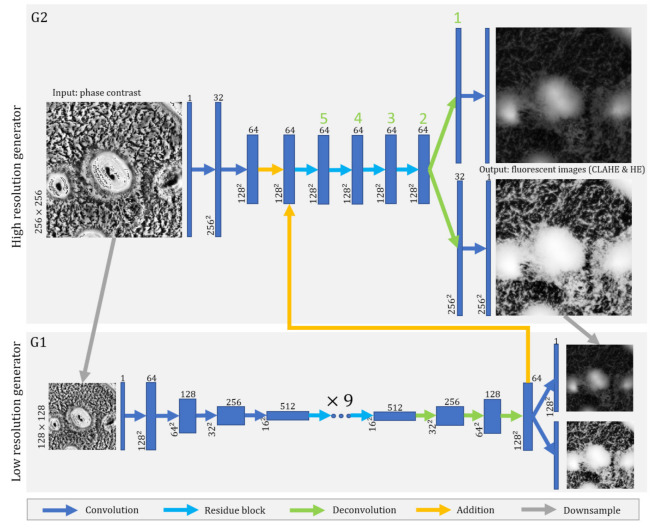
Network architecture of pix2pixHD-HE generator. The lengths and widths of the feature maps are annotated on the left. The depths of the feature maps are annotated as superscripts. The blue dots represent several 2-layer residual building blocks, above which the numbers of blocks are annotated. Each arrow represents an feed-forward operation step. The green numbers label the branch point, and No. 2 branch point produced the best performance (Appendix A). We train the model as described in [28]. We use 2 discriminators to separately discriminate the normalized (CLAHE) and histogram-equalized (HE) fluorescent image channels.

**Figure 3 microorganisms-09-01954-f003:**
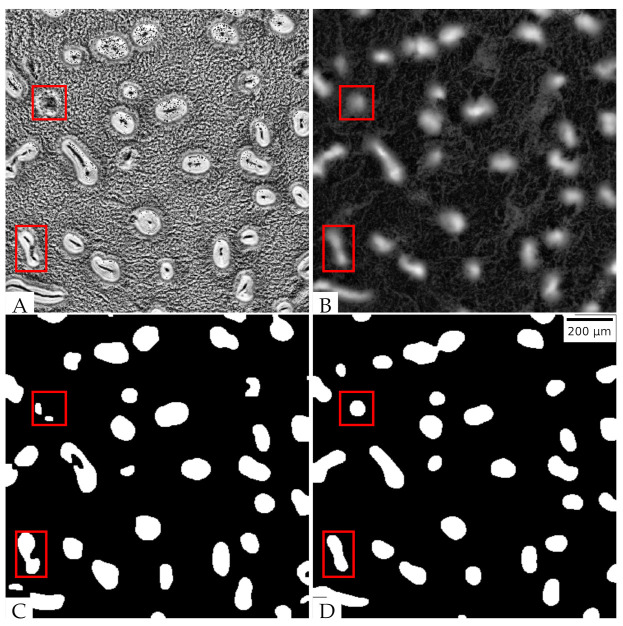
Aggregates extracted from different images at the 740th min: (**A**) Normalized phase-contrast image; (**B**) normalized tdTomato fluorescent image; (**C**) aggregates separated from the phase-contrast image; (**D**) aggregates separated from the fluorescent image.

**Figure 4 microorganisms-09-01954-f004:**
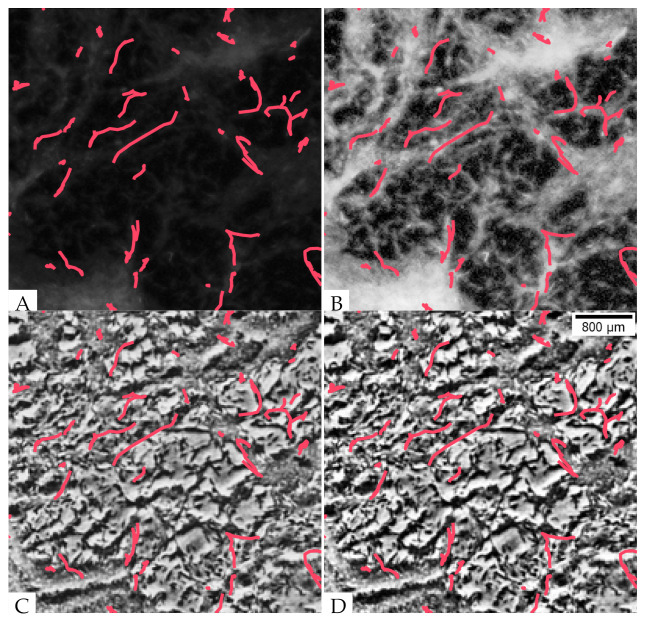
Fluorescent images best represent interaggregate stream patterns. Trajectories of tdTomato-expressing LS3908 cells were tracked over 15 min as they moved through a starving population of GFP-expressing DK10547 cells. Trajectories are superimposed over the corresponding (**A**) GFP fluorescent image, (**B**) histogram-equalized GFP fluorescent image, (**C**) phase-contrast image, and (**D**) histogram-equalized phase-contrast image.

**Figure 5 microorganisms-09-01954-f005:**
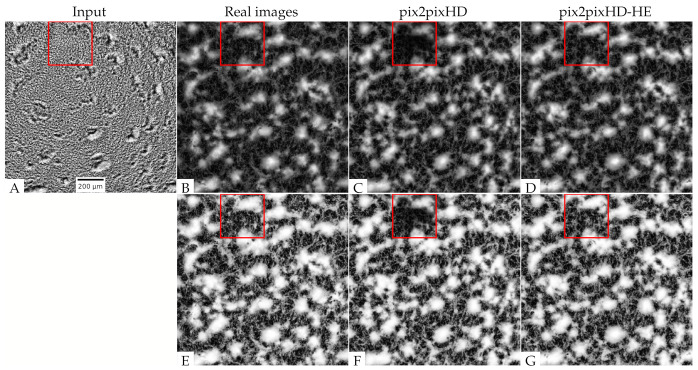
Model performance at the 543 min point in one training movie: (**A**) Input phase-contrast image; (**B**) ground truth tdTomato fluorescent image; (**C**) synthesized fluorescent image (pix2pixHD), the red box indicates a region where the pix2pixHD model fails to accurately reproduce the original image features; (**D**) synthesized fluorescent image (pix2pixHD-HE); (**E**) histogram-equalized ground truth tdTomato fluorescent image; (**F**) histogram-equalized synthesized fluorescent image (pix2pixHD), the red box indicates a region where the pix2pixHD model fails to accurately reproduce the original image features; (**G**) histogram-equalized synthesized fluorescent image (pix2pixHD-HE).

**Figure 6 microorganisms-09-01954-f006:**
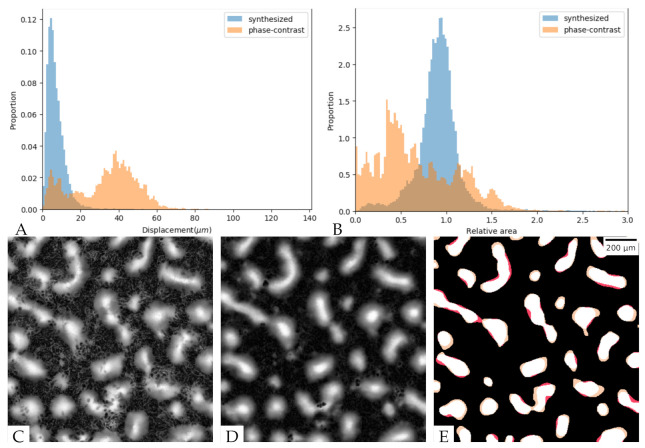
Aggregate segmentation for images of $1\times 10^{10}$ cells/mL cell density. (**A**) The distribution of distances between centroid positions; blue—real and synthesized fluorescent images; orange—phase-contrast and real fluorescent images. (**B**) The distribution of area ratios for the matched aggregate pairs; blue—real and synthesized fluorescent images; orange—phase-contrast and real fluorescent images. (**C**) Fluorescent image at the 800 min. (**D**) Synthesized image at the 800 min. (**E**) Aggregate segmentation result at the 800 min; white—true positive, black—true negative, yellow—false positive, red—false negative.

**Figure 7 microorganisms-09-01954-f007:**
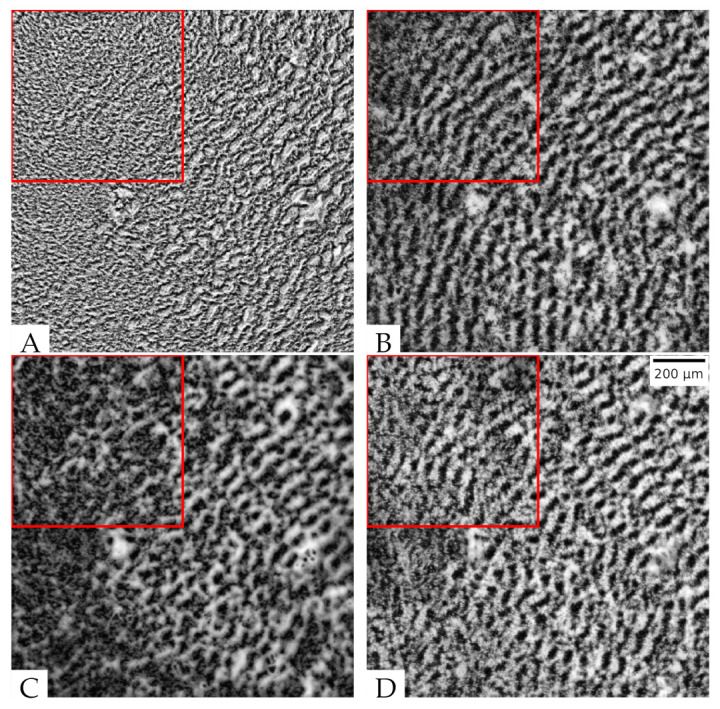
Rippling images and the detected wavelength $\lambda_{k}$ in the red box. The size of the red square box is $\frac{1}{4}$ of the whole image. (**A**) Phase-contrast rippling image $\lambda_{k}∼21$
μm; (**B**) tdTomato fluorescent rippling image $\lambda_{k}=∼46$
μm; (**C**) synthesized fluorescent rippling image before training $\lambda_{k}∼63$
μm; (**D**) synthesized fluorescent rippling image after training $\lambda_{k}∼46$
μm.

**Table 1 microorganisms-09-01954-t001:** Model performance. SD—spatial displacement/μm; σ—minimal standard deviation for added Gaussian noise, given as a percentage of original intensity scale.

			**MSE**				**SSIM**		
**Model**	**Density (cells/mL)**	**Type**	**Value**	**SD**	** σ **	**Difference**	**Value**	**SD**	**Difference**
pix2pixHD	$5\times 10^{9}$	training	0.019 ± 0.014	6.8	6.8%	−0.001 ± 0.009	0.736 ± 0.111	2.2	0.008 ± 0.032
pix2pixHD-HE	$5\times 10^{9}$	training	0.018 ± 0.015	6.2	6.7%		0.744 ± 0.117	2.1	
pix2pixHD	$5\times 10^{9}$	test	0.026 ± 0.015	6.9	8.1%	−0.002 ± 0.008	0.680 ± 0.102	2.2	0.008 ± 0.024
pix2pixHD-HE	$5\times 10^{9}$	test	0.025 ± 0.012	7.0	7.8%		0.687 ± 0.103	2.1	
pix2pixHD	$2.5\times 10^{9}$	training	0.017 ± 0.013	4.1	6.6%	0.000 ± 0.006	0.721 ± 0.128	1.9	−0.001 ± 0.032
pix2pixHD-HE	$2.5\times 10^{9}$	training	0.017 ± 0.011	4.3	6.6%		0.720 ± 0.133	2.0	
pix2pixHD	$2.5\times 10^{9}$	test	0.031 ± 0.022	5.3	8.7%	−0.001 ± 0.008	0.662 ± 0.111	2.2	0.002 ± 0.025
pix2pixHD-HE	$2.5\times 10^{9}$	test	0.030 ± 0.023	4.8	8.6%		0.664 ± 0.103	2.1	
pix2pixHD	$1\times 10^{10}$	test	0.119 ± 0.035	22.6	17.3%	-0.032 ± 0.007	0.420 ± 0.069	4.5	0.044 ± 0.010
pix2pixHD-HE	$1\times 10^{10}$	test	0.087 ± 0.029	16.6	14.8%		0.465 ± 0.074	4.0	

**Table 2 microorganisms-09-01954-t002:** Model performance after histogram equalization. SD—spatial displacement/μm; σ—minimal standard deviation for added Gaussian noise, given as a percentage of original intensity scale.

			**MSE**				**SSIM**		
**Model**	**Density (cells/mL)**	**Type**	**Value**	**SD**	** σ **	**Difference**	**Value**	**SD**	**Difference**
pix2pixHD	$5\times 10^{9}$	training	0.101 ± 0.042	3.3	15.9%	−0.005 ± 0.010	0.458 ± 0.115	2.2	0.019 ± 0.025
pix2pixHD-HE	$5\times 10^{9}$	training	0.096 ± 0.043	3.2	15.5%		0.478 ± 0.123	2.2	
pix2pixHD	$5\times 10^{9}$	test	0.161 ± 0.074	3.7	20.1%	−0.006 ± 0.009	0.385 ± 0.125	2.4	0.016 ± 0.017
pix2pixHD-HE	$5\times 10^{9}$	test	0.155 ± 0.079	3.6	19.7%		0.401 ± 0.131	2.3	
pix2pixHD	$2.5\times 10^{9}$	training	0.098 ± 0.076	2.9	15.7%	−0.003 ± 0.006	0.487 ± 0.134	2.2	0.012 ± 0.015
pix2pixHD-HE	$2.5\times 10^{9}$	training	0.095 ± 0.078	2.7	15.4%		0.499 ± 0.139	2.1	
pix2pixHD	$2.5\times 10^{9}$	test	0.182 ± 0.064	2.5	21.3%	0.003 ± 0.010	0.338 ± 0.107	2.3	−0.001 ± 0.019
pix2pixHD-HE	$2.5\times 10^{9}$	test	0.184 ± 0.066	2.5	21.5%		0.336 ± 0.114	2.3	
pix2pixHD	$1\times 10^{10}$	test	0.123 ± 0.051	5.2	17.6%	−0.001 ± 0.005	0.472 ± 0.098	3.0	0.005 ± 0.008
pix2pixHD-HE	$1\times 10^{10}$	test	0.122 ± 0.049	5.1	17.5%		0.477 ± 0.094	2.9	

**Table 3 microorganisms-09-01954-t003:** Segmentation accuracy for individual aggregates.

Cell Density (cells/mL)	Image Type	Displacement (μm)	Relative Area	Precision	Recall
$2.5\times 10^{9}$	synthesized	4.2 ± 3.1	1.04 ± 0.31	0.82	0.88
$2.5\times 10^{9}$	phase-contrast	10.0 ± 7.3	1.18 ± 0.40	0.69	0.81
$5\times 10^{9}$	synthesized	4.4 ± 3.3	1.18 ± 0.31	0.82	0.95
$5\times 10^{9}$	phase-contrast	9.9 ± 10.6	1.27 ± 0.42	0.70	0.88
$1\times 10^{10}$	synthesized	7.1 ± 5.1	0.90 ± 0.24	0.90	0.83
$1\times 10^{10}$	phase-contrast	32.6 ± 16.6	0.68 ± 0.43	0.48	0.47

## Data Availability

The model is available at https://github.com/IgoshinLab/pix2pixHD-HE.git ((GitHub), accessed on 25 August 2021).

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
