# Peer review of "Quantification of Myxococcus xanthus Aggregation and Rippling Behaviors: Deep-Learning Transformation of Phase-Contrast into Fluorescence Microscopy Images"

_microorganisms, 2021, doi:10.3390/microorganisms9091954_

Round 1

Reviewer 1 Report

With the caveat that I am not an expert in microscopy or deep learning, this study seems to generate a helpful approach to mitigating shortcomings of two standard microscopy approaches, seems to be well performed, including with rigorous model testing, and is generally well written. I think it can be published largely as is.   11 and throughout. I suggest calling the converted images something other than simply fluorescent images, since they are not generated by actual fluorescence. Maybe use something like ‘simulated fluorescence’ consistently throughout the paper, since the authors occasionally qualify with ’synthesized’ vs ‘real’ fluorescence?

32-34. I think this statement is made too strongly, as the results of Zhang et al. 2012 seem to have suggested the hypothesis that “ waves allow M. xanthus cells to quickly cover their prey and remain in place for longer while lysing prey cells and scavenging the resulting nutrients, but not to have conclusively demonstrated its correctness. I thus suggest inserting ‘may’ before ‘waves’.

303. Tables, plural.

Reviewer 2 Report

Myxococcus xanthus multicellular microorganism with different modes of cell organization and communication. It organizes through swarming, a multicellular behavior defined by cells migrating together to efficiently find a prey bacteria. Under starvation, it aggregates into fruiting bodies where some will differentiate into environmentally resistant spores.

This behavior is difficult to track and quantify by phase-contrast microscopy, and fluorescence microscopy is required for quantifying both the aggregates and streams in the inter-aggregate space. In this paper, authors have aimed at the development of an improved image-processing algorithm that transforms phase-contrast microscopy images into fluorescent images. They have used this algorithm to quantify size and morphology of the aggregates from bright field images and they provide really nice results of the quantification. They have also tried to use the same algorism to quantify rippling. After a training process of the pix2pixHD-HE to learn the rippling patterns, they provide clear evidences of an accurate quantification of their wavelength.  

This work provides a step-forward in the quantification of multicellular behavior of Myxococcus xanthus, and I encourage to authors to develop a ease-to-handle program/application to share with the scientific community this algorithm. I also encourage authors to work on the use of this algorithm for the quantification of cellular organization in other microorganisms.
